# Trapped Planetary (Rossby) Waves Observed in the Indian Ocean by Satellite Borne Altimeters

Yair De-Leon[1], Nathan Paldor[1]

[1]Fredy and Nadine Herrmann Institute of Earth Sciences, The Hebrew University of Jerusalem, Edmond J. Safra Campus,
Givat Ram, Jerusalem, 9190401, Israel

*Correspondence to*: Nathan Paldor (nathan.paldor@huji.ac.il)

**Abstract.** Using 20 years of accurately calibrated, high resolution, observations of Sea Surface Height Anomalies (SSHA) by satellite borne altimeters we show that in the Indian Ocean south of the Australian coast the low frequency variations of SSHA are dominated by westward propagating, trapped, i.e. non-harmonic, Rossby (Planetary) waves. Our results demonstrate that the meridional-dependent amplitudes of the SSHA are large only within a few degrees of latitude next to the South-Australian coast while farther in the ocean they are uniformly small. This meridional variation of the SSHA signal is typical of the amplitude structure in the trapped wave theory. The westward propagation speed of the SSHA signals is analyzed by employing three different methods of estimation. Each one of these methods yields speed estimates that can vary widely between adjacent latitudes but the combination of at least two of the three methods yields much smoother variation. The estimates obtained in this manner show that the observed phase speeds at different latitudes exceed the phase speeds of harmonic Rossby (Planetary) waves by 140 % to 200 % (which was also reported in previous studies). In contrast, the theory of trapped Rossby (Planetary) waves in a domain bounded by a wall on its equatorward side yields phase speeds that approximate more closely the observed phase speeds in the study area.

## 1 Introduction

The analysis of observations of Sea Surface Height Anomalies (SSHA), i.e., the deviation of the Sea Surface Height from its mean value at any given point in the ocean, was carried out since the 1990s in various parts of the world ocean by various satellite borne altimeters. Chelton and Schlax (1996), for example, analyzed the first three years of altimetry data collected by the TOPEX/Poseidon satellite in the world ocean, Zang and Wunsch (1999) analyzed five years of TOPEX/Poseidon data in the North Pacific Ocean and Osychny and Cornillon (2004) analyzed six years of modified TOPEX/Poseidon data in the North Atlantic Ocean. Additional observational studies are summarized in Barron et al. (2009) and references therein.

In most parts of the ocean the satellite observations showed a ubiquitous and pronounced westward migration of SSHA with amplitude of a few centimetres. This westward, rather than eastward, propagation led to the interpretation of these observations as a surface manifestation of the first baroclinic mode of planetary (also known as Rossby) waves that

propagate westward (i.e. their phase speed is negative) in the ocean thermocline. Recent studies (e.g., Chelton et al., 2007; Chelton et al., 2011), however, argue that the observed SSHA features belong to mesoscale eddies and are not surface manifestations of planetary waves in the thermocline but this change of view has no effect on the estimate of the westward propagation speed since these eddies propagate westward at the same phase speed as of long Rossby waves (Chelton et al., 2011; O'Brien et al., 2013; Polito and Sato, 2015; see also Nof, 1981 for theoretical estimate of eddy migration rate on the $\beta$-plane).

The quantification of the rate of westward propagation of the observed SSHA features is based on the construction of time-longitude (also known as Hovmöller) diagrams at a given latitude. The slopes of contours on these diagrams are proportional to the propagation speed of the SSHA features. These slopes can be calculated using methods that are commonly employed in image processing such as the Radon transform (or its more recent alternative – the variance method) and the Two Dimensional Fast Fourier Transform (2D FFT) which are described in details in Sect. 2.2 below.

Previous studies of the westward propagation of observed SSHA in mid-latitudes have all yielded rates of westward propagation that are faster than the phase speeds predicted by the harmonic planetary wave theory (see below for details). Explanations for these underestimates by the harmonic theory were proposed which are based on considerations that involve either the addition of mean zonal flows in the equations (Killworth el al., 1997 and see also Colin de Verdière and Tailleux, 2005, who emphasized the curvature effect of the mean flow rather the mean flow itself) or the influence of the bottom topography (Tailleux and McWilliams, 2001) while Killworth and Blundell (2005) applied a combination of these two effects. Watanabe et al. (2016) showed that the standard linear wave theory can be tailored to fit the observations in the tropics by considering parameters such as effective-$\beta$ (that includes the meridional gradient of the background potential vorticity) and forcing by Ekman pumping. LaCasce and Pedlosky (2004) argued that due to baroclinic instability the wave structure is changed and becomes more barotropic so it propagates faster and no mean flow is required. Along similar lines, Hochet et al. (2015) suggested that the assumption that observations are of the first baroclinic mode cannot be made a-priori, but the vertical structure is predicted from the altimetry data. Thus, they found that in some regions the vertical structure is more barotropic than baroclinic so the theoretical phase speed is larger and no discrepancy exists between theory and observations. By incorporating physical elements that are not included in the simple linear wave theory of the Shallow Water Equations (e.g. velocity shear, non-linear terms, topography, mean flows and juxtaposing barotropic and baroclinic modes) these (and other) past studies were successful in bridging some of the discrepancies found between the observed SSHA propagation speeds and the phase speeds of harmonic wave theory.

In contrast to the phase speed, other wave characteristics such as the meridional variations of the SSHA amplitudes (which are predicted by the harmonic theory to be sinusoidal) have never been verified in these past studies. The reason is that in the framework of the harmonic theory (see more details below) the central latitude, $\phi_0$, which determines the origin of the $y$- (meridional) coordinate, is determined by the latitude of observation. Thus, observations of SSHA at adjacent latitudes

cannot be compared to one another since their y-dependencies are determined by the same equations but with different origins so the same y-coordinate denotes different points in the two sets of equations.

The interpretation of these SSHA observations has employed the harmonic theory of westward propagating, low frequency, waves that assumes the existence of a zonal channel that bounds the north-south extent on the $\beta$-plane. Under these assumptions zonally propagating wave solutions of the Shallow Water Equations can be constructed and explicit expressions can be derived for both the zonal phase speed of the waves and the spatial structure of their amplitudes. The emerging spatial structure of the waves is oscillatory (harmonic) in both the zonal and meridional coordinates i.e., the waves simply oscillate with wavenumber $k$ in the zonal direction and wavenumber $l$ in the meridional direction (Pedlosky, 1982; Cushman-Roisin, 1994; Vallis, 2006).

An alternative to the traditional harmonic theory is the trapped wave theory which was developed on the mid-latitude $\beta$-plane by Paldor et al. (2007) and Paldor and Sigalov (2008). In this theory the meridional variation of the wave's amplitude is not harmonic but is given instead by the Airy function (see details in Sect. 4.1 below) and the requirement of two channel walls of the harmonic theory is replaced in this trapped wave theory by a single wall that marks the equatorward boundary of the domain. In sufficiently wide meridional ranges the phase speed of the trapped waves is higher than that of the corresponding harmonic waves by a factor of 2 to 4.

The current study employs the available series of SSHA observations sampled on a 1/4° spatial grid which are compared to the theoretical phase speeds and meridional structures of the height field using the trapped, and harmonic, wave theories. The comparisons provide a measure of the relevance of the trapped and harmonic wave theories to the observed SSHA fields in the Indian Ocean.

This paper is organized as follows: Section 2 provides details of the observations and methods used for estimating the observed phase speed and in Sect. 3 we compare the theoretical and observational meridional variation of the height field in the Indian Ocean south of the Australian coast (which includes the Great Australian Bight). Section 4 describes theoretical expressions for the phase speed and the meridional structure of the height field of the harmonic and trapped wave theories that are compared with SSHA observations in the region of interest in Sect .5. The paper ends in Sect. 6 with summary and discussion of the findings.

## 2 Data and Methods

### 2.1 SSHA Data

The altimetry products used for a comparison with theory were produced by Ssalto/*Duacs* and distributed by *Aviso*, with support from *CNES*. The data we used are the multi-mission (i.e., up to four satellites at a given time, e.g., TOPEX/Poseidon, Jason 1, Jason 2, Envisat) gridded Sea Surface Heights, sampled on a 1/4°×1/4° Cartesian grid once a week from 1/1/1993 to

31/12/2012. These data are improved compared to those used in previous studies since the combination of data from several, present-day, satellites enables high precision altimetry in both time and space at finer resolutions. More details on the way the SSHA data are produced by *Aviso* can be found at http://www.aviso.altimetry.fr/duacs/. The SSHA time-series of each grid-point in this region were low-pass filtered in the present study by performing a 5-week running-average to eliminate short-term variability such as storms, tides (including the fortnightly component) and other variations of periods less than one month. Though this filtering leaves parts of the high frequency signals in the averaged signal these parts are minute since the window contains many cycles of the high amplitude signals such as the M2 tides. Calculations with wider windows of 27 and 53 weeks (done to examine the possible contribution of longer term variability such as seasonal winds) yielded qualitatively identical results (see details in Sect. 3 below).

## 2.2 Methods of estimating observed phase speed of SSHA

The basis for estimating the speed of westward propagation of SSHA is time-longitude (Hovmöller) diagrams of the SSHA field at fixed latitude. In this diagram the westward propagation is evident from the left-upward tilt of constant SSHA values i.e., same color contours and the angle between this tilt and the ordinate is directly proportional to the speed of westward propagation. The diagram provides a time-series of the SSHA changes at fixed longitude and a longitude variation series at any particular time so Fast Fourier Transforms can be easily calculated in time and longitude to yield the frequency and zonal wavenumber spectra of observed SSHA.

Three objective methods are employed in the literature for calculating the phase speed of waves from time-longitude diagrams. The first method is the frequently used (e.g., Chelton and Schlax, 1996; Chelton et al., 2003; Tulloch et al., 2009) Radon transform used in image processing for detecting structures on any digital image (see details in e.g., Jain, 1989). The Radon transform of a two dimensional function $f(x, y)$ that describes the intensity of an image at $(x, y)$, such as SSHA values in a given (longitude, time) domain, is the integral of $f(x, y)$ along a line $L$ inclined at an angle $\theta$ relative to the ordinate (i.e., $\theta \pm 90°$ relative to the abscissa) and displaced a distance $s$ from the origin. For each angle $\theta$ we sum the squares of the values of the integrals along all lines having the same $\theta$ (i.e., having different distance $s$). The angle at which this sum-of-squares attains its maximum is the most accurate estimate for the orientation of structures with the same SSHA value on the time-longitude diagram. The tangent of this preferred $\theta$ is proportional to the sought westward propagation speed. Note that in order to minimize the effect of few very high entries on the sum-of-squares we apply the Radon transform to a modified time-longitude diagram where the signal is scaled on the [0,1] interval and the mean of the scaled signal is subtracted. The second method is a relatively new algorithm (Polito and Liu, 2003; Barron et al., 2009) that constitutes an adaptation of Radon transform to a propagating wave. In this method the **variance** of amplitude values is calculated along the same lines. For each angle $\theta$ we average the variances along all lines at allowed distances $s$ and the westward propagation speed is then determined by the tangent of the angle $\theta$ at which the mean of variances is minimal. The third method commonly used (e.g.,

Zang and Wunsch, 1999; Osychny and Cornillon, 2004) to obtain the observed phase speed is the application of the 2D FFT to the time-longitude diagram to get a frequency-wavenumber (i.e., $\omega$, $k$) diagram of the signal's amplitude. The phase speed is obtained by locating the values of $\omega$ and $k$ where the amplitude is maximal (i.e., maximum spectral coefficient) and calculating $C=\omega/k$ at this point of maximum spectral coefficient. Alternatively, the directionality of the spectral coefficients

in the ($\omega$, $k$) diagram can be found by sweeping over all lines that pass through the origin and inclined at angles ranging from $0°$ to $180°$ relative to the abscissa (i.e., $\omega/k$ lines). The value of $C$ is then determined as the slope of the line of maximal sum-of-squares of spectral coefficients ("total energy").

A comparison between the three methods was made using synthetic signals (De-Leon and Paldor, 2016). Based on the insight gained from the study of synthetic signals, an estimation of the observed phase speed is accepted here only when an

isolated peak (characterized by the point at which the derivative changes sign at a clearly defined sharp peak and maintains the same sign in bands that are at least $3°$ wide on either side of the peak) is evident in at least two of the three methods and the phase speeds that correspond to these peaks agree by better than 10 %.

Note that the observed phase speed is obtained from the Hovmöller diagrams in units of one-quarter degree longitude per week which is converted to units of one centimetre per second by multiplying the observed phase speed by $4.6\cos\phi_m$

(where $\phi_m$ is the latitude of observation).

## 2.3 The study domain in the Indian Ocean

The trapped wave theory in mid-latitudes applies without any modification to domains of large meridional extent (so the $\beta$–plane approximation applies) that are bounded on their equatorward side by a wide zonal boundary. As shown in Fig. 1 such a nearly zonal boundary exists in the Indian Ocean south of Australia. The domain of study extends from the south coast of

Australia at about $31.5°$ S to only about $45°$ S since south of this latitude the SSHA field is strongly affected by the nearly 2000 km wide, fast, and strongly meandering, Antarctic Circumpolar Current (ACC).

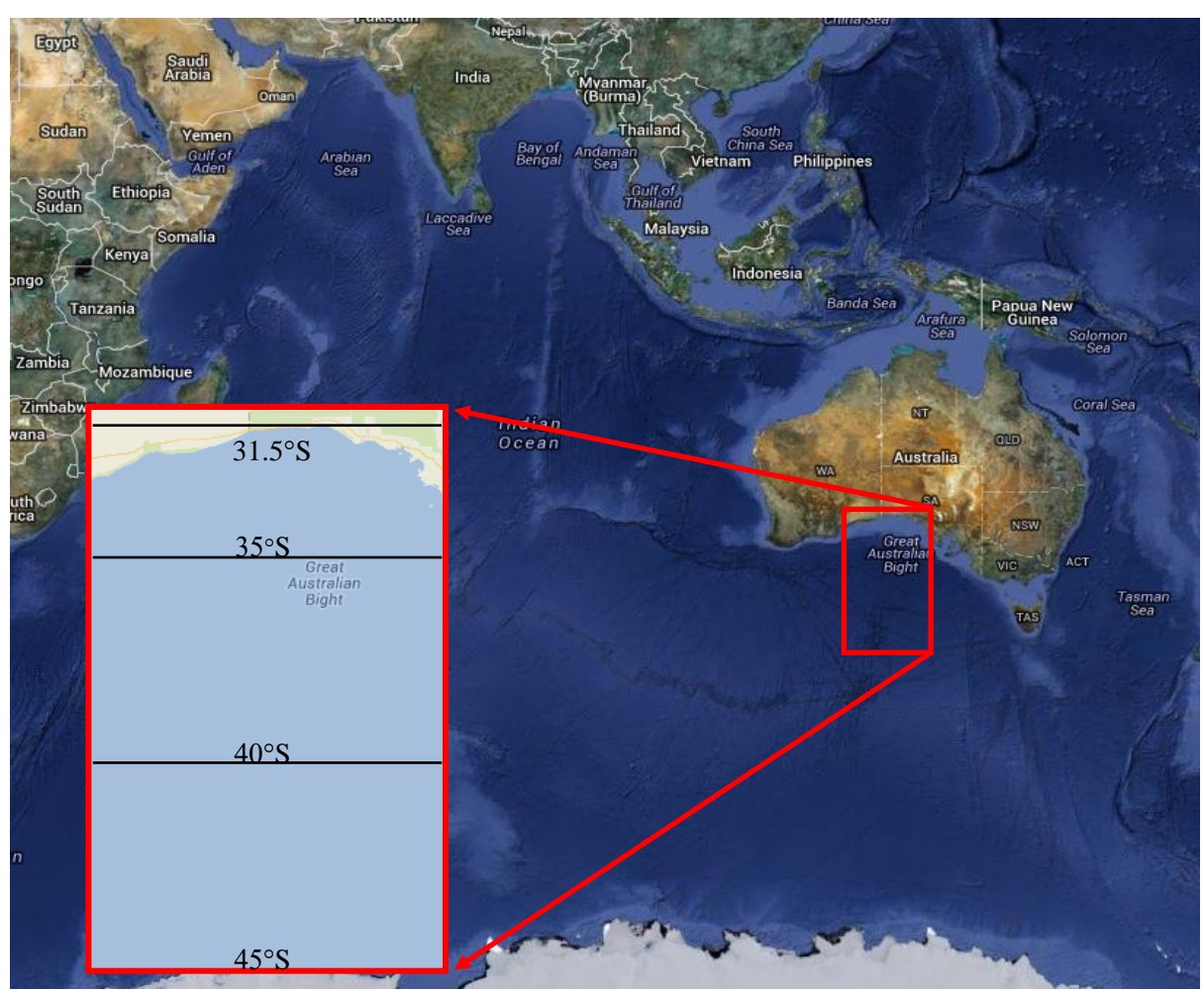

**Figure 1: The domain under study in the Indian Ocean with a zoom in on the longitude band of 124.5° E - 134.5° E where altimetry data are analyzed (reproduced from Google Maps).**

### 3 The Meridional structure of SSHA

5    The standard deviation of the temporal changes of SSHA observations in each point of this domain over the entire 20 years is shown in Fig. 2(a), which clearly demonstrates an increase in the SSHA signal from 4 cm to 9 cm in the band of width 2°-3° right next to the Australian coast. The meridional structure of the observed SSHA is clearly non-uniform, while in the harmonic (oscillatory) theory the height field is uniform (i.e., constant) for $l$=0 and sinusoidal for $l$>0. Although the ocean depth decreases towards the shore, the observed variability of SSHA signal there cannot be attributed to topography since

10    steady winds affect only the average displacement of the sea surface which is subtracted from the SSHA signal when the standard deviation is calculated, while the effect of winds of periods shorter than 5 weeks are filtered out by our low-pass

filter. In order to examine the possible effect of longer term winds (seasonal to annual) the calculations were repeated with windows of 27 weeks and 53 weeks. These calculations yielded very similar results to those obtained with the 5 weeks window but with slight decrease in the amplitude of main signal near the coast and minute changes in the structure far from it. For the same reason this coastal peak cannot be associated with a mean long-coast current since such a current will not show up on a map of temporal standard-deviation. The 5 week filter which we applied to the data also eliminates high-frequency waves such as Kelvin waves and topographic Rossby waves (or continental shelf waves) since in the Great Australian Bight where the slope is 0.01 the period of these waves is O(1 day) (see e.g. Cushman-Roisin, 1994 for harmonic waves and Cohen et al., 2010 for non-harmonic waves).

The mean over all longitudes (Fig. 2(b), thick blue line) of the 41 individual latitudinal cross-sections (thin light-blue lines) is compared with analytical expressions (described in the next section) for the meridional structure of the height field of both the trapped wave theory (dashed red line) and the harmonic theory (dotted green line). The decay rates with latitude of both observed and trapped wave theoretical curves are similar in contrast to the flat curve of the harmonic theory. An unexplained minor secondary peak is found near 36° on the observed curve (also evident near 36° S, 125° E in panel (a)) and upon examining a larger SSHA map it turns out that this secondary peak is an eastward extension of the Leeuwin current that flows poleward along the west coast of Australia between March and July (Godfrey and Ridgway, 1985). Alternatively, this peak can be interpreted as a poleward propagation (into the Indian Ocean) of energy generated in the equatorial Pacific Ocean by the wind and by Ekman pumping which forms Rossby waves in the study area (Potemra, 2001).

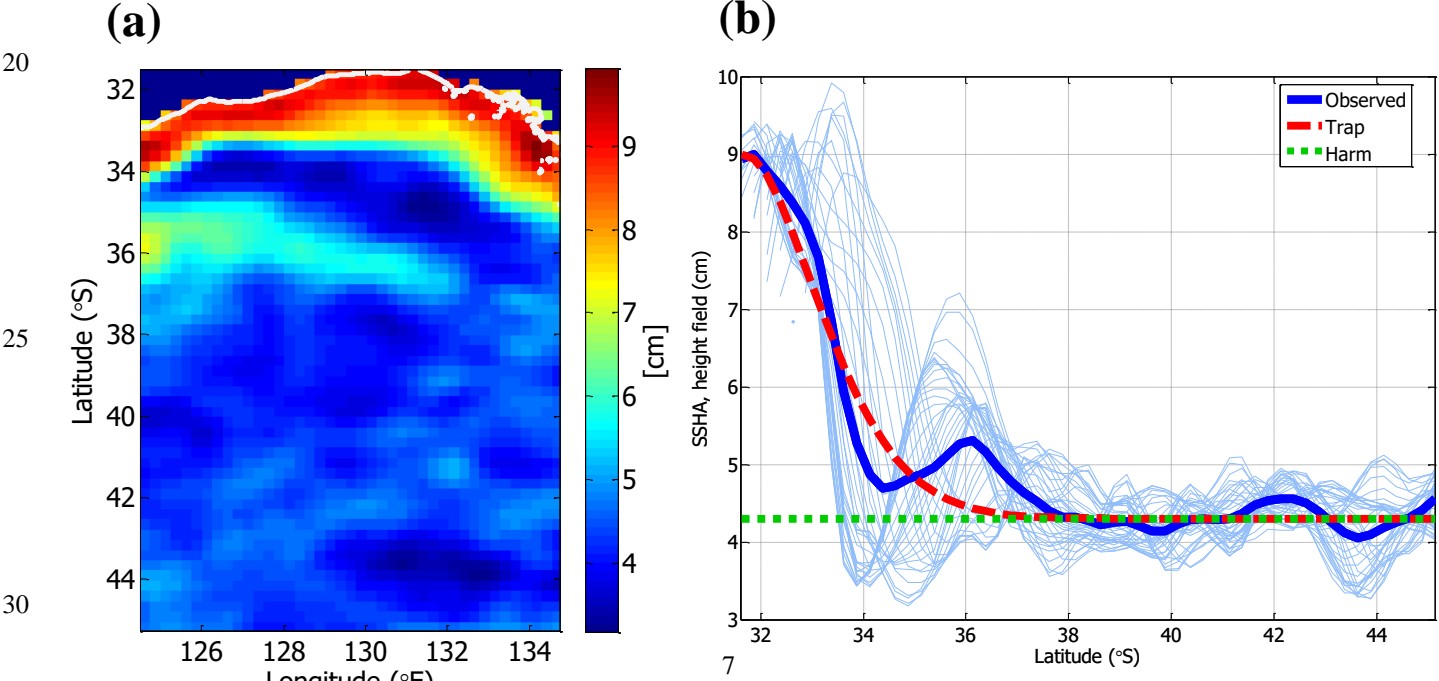

**Figure 2. (a) The temporal standard deviation of satellite derived SSHA over the entire 20 year period poleward of the Great Australian Bight. The coastline is plotted in white. (b) Latitudinal cross-sections of the data of 2(a) every 0.25 degree longitude (thin light-blue lines), the mean over all longitudes of the latitudinal cross-sections (thick blue line) and the analytical expression for the meridional structure of the height field of trapped waves for zero zonal wavenumber and zero meridional mode number (dashed red line, see Eq. 5 below). The maximal trapped wave amplitude is set to match that of the mean observed cross-section where the off-shore minimum is about 4 cm since the temporal mean of *Aviso*'s original data is not zero. The analytical expression for the meridional structure of the height field of harmonic waves for k=0 and l=0 is constant i.e. described by a straight line parallel to the abscissa at arbitrary value of the ordinate, here it is set to match the off-shore minimum of about 4 cm (dotted green line).**

## 4 Application of wave theories to observations

The relevance of the trapped wave theory to observations can be best assessed by comparing the theoretical phase speed and meridional structure of the waves with observations such as those described above. In addition, it is also natural to compare the observations with phase speed and meridional structure of the harmonic planar theory and use the observations to assess the applicability of each of these theories.

### 4.1 Explicit expressions for the phase speeds of the two wave types

The Coriolis frequency on the $\beta$-plane, expanded linearly about some latitude, $\phi_0$, is given by $f(y) = f_0 + \beta y = 2\Omega \sin \phi_0 + \frac{2\Omega}{a} \cos \phi_0 \cdot y$, where $\Omega$ is the frequency of Earth's rotation about its polar axis, $a$ is Earth's radius and $y = (\phi - \phi_0) \cdot a$ (where $\phi$ is the latitude) is the north coordinate.

In a channel on the mid-latitudes $\beta$-plane where the Coriolis frequency is expanded near $\phi_0 = \phi_m$, the latitude of observation, the fastest baroclinic phase speed (in units of metre per second) of harmonic Rossby waves is (see Cushman-Roisin, 1994; Vallis, 2006):

$$C^{\text{harm}} = \frac{-\beta}{k^2 + l^2 + \frac{f_0^2}{g'H'}} = \frac{-\frac{2\Omega}{a} \cos \phi_m}{k^2 + l^2 + \frac{(2\Omega)^2 \sin^2 \phi_m}{g'H'}}, \tag{1}$$

where $k$ and $l$ (the latter is denoted in other studies by $n$) are the zonal and meridional wavenumbers of the Cartesian coordinates, respectively, $g'$ is the reduced gravity and $H'$ is the weighted depth of the two (or more) layers that make up the baroclinic ocean so $(g'H')^{1/2}$ is the speed of gravity waves. For sufficiently long waves when both $k$ and $l$ can be neglected this phase speed reduces to:

$$C^{\text{harm}} = \frac{-\beta}{\frac{f_0^2}{g'H'}} = \frac{-g'H' \cos \phi_m}{2\Omega a \sin^2 \phi_m}. \tag{2}$$

In contrast to the harmonic wave theory which is fully described in many textbooks the application of the trapped wave theory requires some more detailed explanation. In this theory, the waves are trapped next to a single wall that marks the

equatorward boundary of the domain and the meridional variation of the wave's amplitude is given by the regular (at infinity) Airy function, $Ai$, that oscillates (but is not periodic in contrast to harmonic/sinusoidal oscillations) in the $(-\infty, 0)$ interval and decays to zero faster than exponential in the $(0, +\infty)$ interval (see e.g., Abramowitz and Stegun, 1972). The phase speed of trapped waves in a mid-latitude channel is (see Eq. (6) of Gildor et al. (2016)):

$$C^{\text{trap}} = \frac{-\beta}{k^2 + \frac{f_0^2}{g'H'} + \zeta_n \cdot \left(\frac{2f_0\beta}{g'H'}\right)^{2/3} - \frac{2f_0\beta}{g'H'} \cdot y_w}, \tag{3}$$

5 where $\zeta_n$ is the absolute value of the $n$th zero of $Ai$ and $y_w$ is the location of the equatorward wall. Following the studies of Paldor and Sigalov (2008) and De-Leon and Paldor (2009), we expand here the Coriolis frequency near $\phi_0 = \phi_w$, where $\phi_w$ is the latitude of the equatorward boundary of the domain so $y_w = 0$ there in which case the last term in the denominator of Eq. (3) vanishes (in contrast to Gildor et al. (2016) where the wall was placed at $y_w=-L/2$ where $L$ is the channel width). In addition, the boundary condition at $\phi_w$ in Gildor et al. (2016) is the vanishing of the meridional velocity while in the present 10 application Fig. 2(b) implies that the meridional derivative of the height field vanishes at $\phi_w$. Thus, $\zeta_n$ in Eq. (3) should be replaced in the present application by $\xi_n$ - the absolute value of the $n^{\text{th}}$ zero of the derivative of $Ai$ (see the discussion following Eq. (5) below). The resulting expression for the phase speed of the first baroclinic mode of sufficiently long trapped waves (i.e., for zonal wavenumber $k=0$ and meridional mode number $n=0$ for which $\xi_0=1.0188$, see P. 478 of Abramowitz and Stegun, 1972) is:

$$C^{\text{trap}} = \frac{-\beta}{\frac{f_0^2}{g'H'} + 1.0188 \cdot \left(\frac{2f_0\beta}{g'H'}\right)^{2/3}} = \frac{-\frac{2\Omega}{a}\cos\phi_w}{\frac{(2\Omega)^2\sin^2\phi_w}{g'H'} + 1.0188 \cdot \left(\frac{(2\Omega)^2}{ag'H'}\sin 2\phi_w\right)^{2/3}}. \tag{4}$$

15 Note that in contrast to the planar harmonic theory where $l$ is a meridional wavenumber (measured in units of m$^{-1}$) which cannot be determined when no channel exists (and the same is true for the zonal wavenumber, $k$), in the trapped wave theory $n$ is a non-dimensional mode number that counts the number of zeros of the eigenfunction inside the meridional domain.

The trapped wave theory is valid when the meridional range is larger than $(2 + \zeta_n)\left(\frac{ag'H'}{4\Omega^2\sin 2\phi_w}\right)^{1/3}$ (see Eq. (7) of Gildor et al., 2016). For $n=0$, typical values of $(g'H')^{1/2}$ of 2 to 3 m s$^{-1}$ and $\phi_w=30°$ this condition is satisfied when the domain 20 is wider than about 500 kilometres. Accordingly, the harmonic theory applies only in unrealistically narrow channels that are only a few hundred kilometres wide (see also Fig. 3 in Paldor and Sigalov, 2008).

## 4.2 Explicit expressions for the meridional structure of the two wave types

The meridional structure of the height field of harmonic waves in mid-latitudes varies with $y$, the meridional coordinate, as $A\cos(ly+\Gamma)$ (where $A$ is an arbitrary amplitude and $\Gamma$ is a phase angle that guarantees, together with $l$, that the wave satisfies the boundary conditions) which for $l=0$ yields height and velocity fields that do not vary with $y$.

The meridional structure of the height field of trapped waves is (see Eq. (5) in Gildor et al., 2016 with the modifications outlined in Sect. 4.1):

$$\eta(y) = \frac{H'av_0}{g'H' - C^2}\left\{ C \cdot \left(\frac{4\Omega^2}{ag'H'}\sin 2\phi_w\right)^{1/3} \cdot Ai'\left(\left(\frac{4\Omega^2}{ag'H'}\sin 2\phi_w\right)^{1/3} \cdot y - \xi_n\right)\right.$$

$$\left. - f(y)Ai\left(\left(\frac{4\Omega^2}{ag'H'}\sin 2\phi_w\right)^{1/3} \cdot y - \xi_n\right)\right\}, \tag{5}$$

where $v_0$ is an arbitrary amplitude and the phase speed, $C$, is given by $C^{\text{trap}}$ of Eq. (3) with the modifications outlined in Sect. 4.1. Note that this theoretical expression for $\eta(y)$ consists of two terms: $Ai(y)$ and $Ai'(y)$, so $d\eta/dy$ contains terms proportional to $Ai(y)$, $Ai'(y)$ and $Ai''(y)$. The Airy differential equations relates $Ai''(y)$ to $y \cdot Ai(y)$ so at $y=0$ the $Ai''(y)$ term vanishes and the coefficient of $Ai(y)$ is negligible compared to that of $Ai'(y)$ which clarifies why the extremum of $\eta$ occurs at $y\approx0$.

## 5 Results and comparison between observations and theories

### 5.1 Meridional structure of the height field

The meridional structure of the height field of the trapped waves curve in the area of study is computed from $\eta(y)$ of Eq. (5) with $C = C^{\text{trap}}$ of Eq. (4). In the calculation of these expressions of $C^{\text{trap}}$ and $\eta(y)$ the value of $\phi_w$ was set to 31.5° S, $k=0=n$ (so $\xi_n = \xi_0 = 1.0188$), and $(g'H')^{1/2}$, the speed of gravity waves, was set to 2.8 m s$^{-1}$ following Fig. 2 in Chelton et al. (1998) (see also http://www-po.coas.oregonstate.edu/research/po/research/rossby_radius/). The analytical expression for the meridional structure of the height field of harmonic waves for $k=0$ and $l=0$ is constant i.e. described by a straight line parallel to the abscissa at arbitrary value of the ordinate. As shown in Fig. 2(b) the curve of the trapped wave theory (dashed red line) fits the observed one (solid blue line) much better than that of the harmonic theory (dotted green line).

### 5.2 Phase speeds

An estimation of the speed of westward propagation of observed SSHA is obtained by analyzing time-longitude (Hovmöller) diagrams of the SSHA field at fixed latitude as explained in Sect. 2.2. Figure 3 shows two examples of such diagrams calculated at 36° S (panel a) and at 45° S (panel b); both are sufficiently far from any major current or continent and

sufficiently far from the equatorward boundary so that the condition for the validity of the trapped wave theory derived in the paragraph following Eq. (4) is satisfied (and sufficiently far (at least 200km) from the ACC). Also plotted on these diagrams are the two lines corresponding to the theoretical phase speeds for $k=0$ and $n=0$ of trapped wave theory (Eq. (4), dashed) and the harmonic wave theory (Eq. (2), dotted). A casual visual inspection shows that the line of trapped wave theory fits the observed tilt of SSHA features more closely than the harmonic one (especially at 45° S (panel b)).

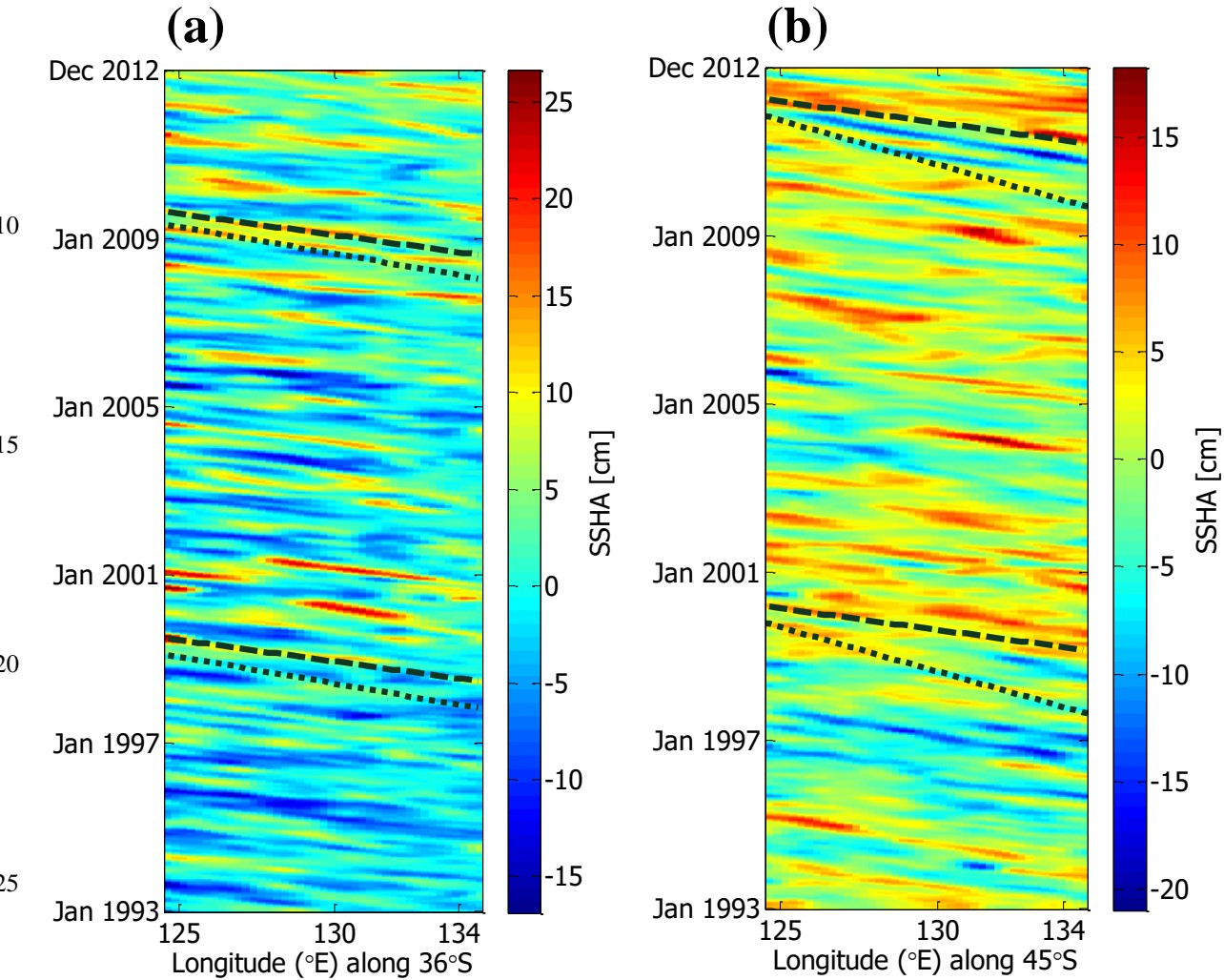

**Figure 3. Time-longitude (Hovmöller) diagrams at $\phi_m$=36° S (panel a) and at $\phi_m$=45° S (panel b), where the abscissa is longitude and the ordinate is the date. The temporal average was subtracted from the record of each grid point. Dashed lines: trapped wave phase speed (Eq. (4)); Dotted lines: harmonic wave phase speed (Eq. (2)).**

The various objective methods for obtaining the phase speed from Hovmöller diagram are now applied to the diagram in Fig. 3(b). The distribution of the sum-of-squares (or standard deviation) of the Radon transform as a function of the angle $\theta$ (for $\theta$ values near the peak) is shown in Fig. 4(a) (solid blue curve) where the maximum is at $\theta \approx 30°$ (i.e., $C \approx 1.9$ cm s$^{-1}$, in absolute value, hereafter). The distribution of the mean of variances as a function of the angle $\theta$ is shown in Fig. 4(b) where the mean of variances is minimal at $\theta = 33°$ (i.e., $C \approx 2.1$ cm s$^{-1}$). The $\theta$-values corresponding to the phase speed of trapped waves (obtained from Eq. (4), $\theta \approx 37°$ i.e., $C \approx 2.4$ cm s$^{-1}$; solid red vertical line) and to the harmonic phase speed (obtained from Eq. (2), $\theta \approx 20°$ i.e., $C \approx 1.2$ cm s$^{-1}$; dashed green vertical line) are also shown in panels (a), (b) and (d) of Fig. 4.

The frequency-wavenumber diagram obtained by applying 2D FFT to the time-longitude diagram at this latitude is shown in Fig. 4(c) in the range of low frequency and low wavenumber (in the rest of the frequency-wavenumber plane the amplitudes vanish). The maximum amplitude (outside $k=0$ since only $k \neq 0$ values yield finite westward phase speeds by $\omega/k$) of the frequency-wavenumber diagram shown in Fig. 4(c) occurs at $k=0.1571$ which is a sufficiently small value that justifies the long-wave approximation made earlier ($k=0.1571$ corresponds to wavelength of about 160 degrees of longitude). The frequency with maximal spectral amplitude at this wavenumber is -0.09045 so the resulting phase speed of maximal spectral amplitude is -0.09045/0.1571=-0.5757 (in degrees of longitude per 4 weeks, i.e., $C \approx 1.9$ cm s$^{-1}$) and this phase speed equals the phase speed obtained independently by the Radon transform. Figure 4(c) also compares the phase speeds of the two theories with the observed speed and it demonstrates that the phase speed of trapped waves (dashed red line) is slightly (but not significantly) closer to the observed speed (defined by both the maximum amplitudes and the directionality of the band of high amplitudes in frequency-wavenumber plane) than that of the harmonic waves (dotted light-green line). Though this red line (that corresponds to trapped waves) connects the two maximal values of the 2D FFT at the smallest $\pm k \neq 0$ (and passes through the origin as expected), at larger $k$ its fit to the location of maximal amplitude is no better than that of the line corresponding to harmonic waves.

The distribution of the sum-of-squares of the spectral coefficients along $\omega/k$ lines versus the inclination angle, arctan($C$), is shown in Fig. 4(d) (blue curve) where the curve attains its maximum at arctan($C$) $\approx 151°$ i.e. $\theta \approx 29°$ in terms of the Radon transform method ($C \approx 1.8$ cm s$^{-1}$).

For this time-longitude diagram the phase speed obtained by the variance method differs by about 11 % to 14 % from that obtained by the Radon and 2D FFT methods that yield nearly identical phase speeds, so according to our criteria mentioned in the end of Sect. 2.2 the latter estimate for the phase speed is accepted. However, this observed phase speed does not clearly validate any of the two theoretical phase speeds since the corresponding vertical lines in panels (a) and (d) of Fig. 4 are located at nearly the same distance on both sides of the observed peak. In contrast, the estimate of the observed phase speed obtained by the variance method (Fig. 4(b)) is much closer to that of the trapped wave phase speed than the harmonic one. Thus, the determination of the relevant theory that yields the correct phase speed that matches the propagation

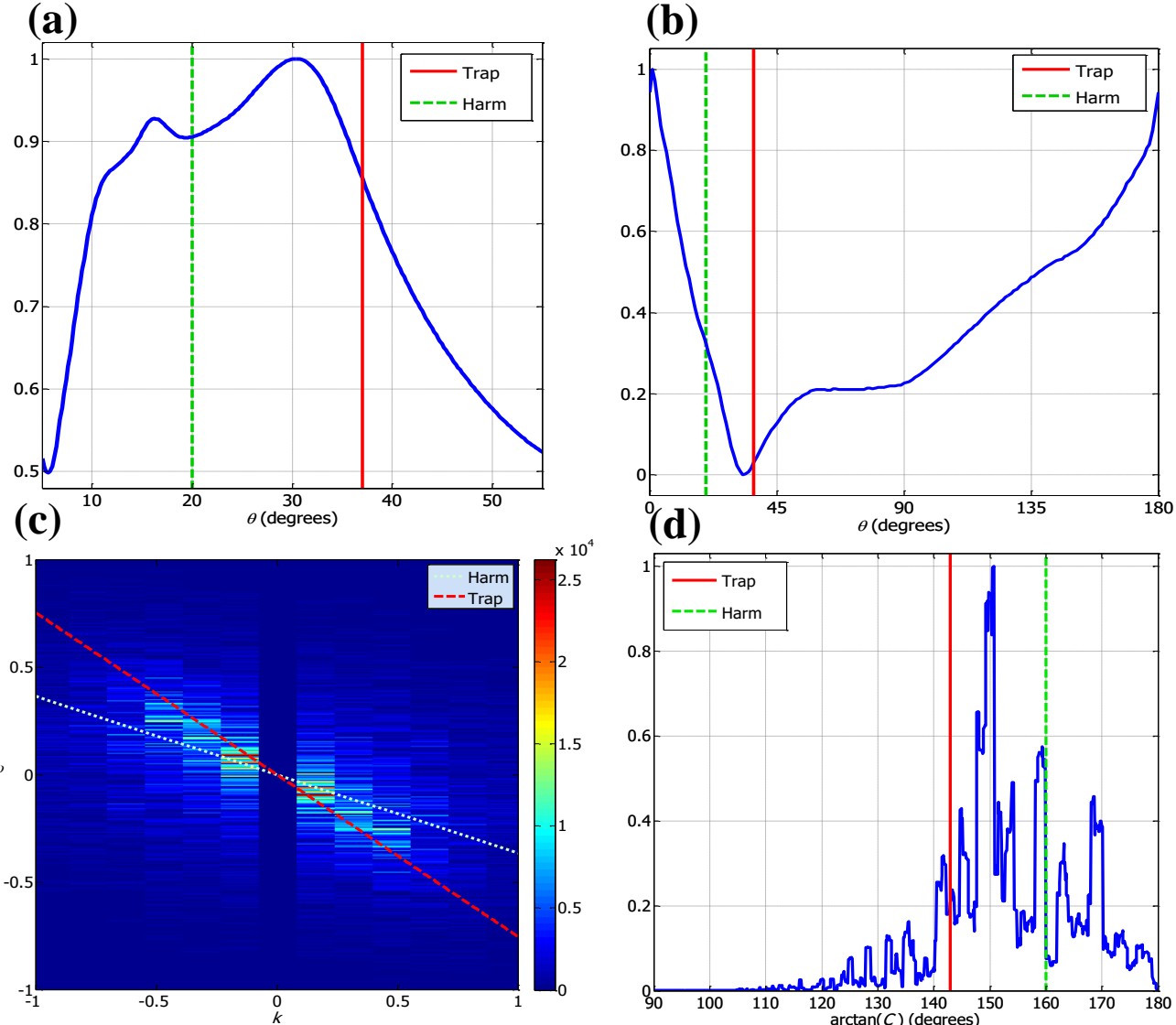

**Figure 4. Analyses of the phase speeds of the Hovmöller diagram of Fig. 3(b). (a) Solid blue curve: The sum-of-squares of the Radon transform as a function of $\theta$ (near the peak) normalized such that the maximum value equals 1; Dashed green vertical line: the angle of the harmonic wave theory (Eq. (2); Solid red vertical line: the angle of the trapped wave theory (Eq. (4);). The same two vertical lines appear also in panels (b) and (d). (b) Solid blue curve: The distribution of the mean of variances versus $\theta$, normalized such that the maximum (maximum) values equal 1 ( 0). (c) The 2D FFT frequency-wavenumber diagram in the low frequency-low wavenumber regime ($k$ is measured in units of (¼ degrees of longitude)$^{-1}$, $\omega$ is measured in units of week$^{-1}$ and the amplitude units are arbitrary). Dashed red line: trapped wave's phase speed, Eq. (4); Dotted light-green line: harmonic wave's phase speed, Eq. (2). (d) The distribution of the sum-of-squares of the 2D FFT amplitudes along different lines (sweeping) versus arctan($C$), normalized such that the maximum value equals 1 (blue curve). Only values of $90° <$ arctan($C$) $< 180°$ are shown since only these values correspond to westward propagating speeds.**

rate determined from observations cannot rely solely on the match at any particular latitude and therefore match over an entire range of latitudes was also examined.

The implications from similar comparisons carried out every 0.5° between 33° S and 45.5° S can be summarized as follows: At about third of the diagrams analyzed the signal was too blurred or the three methods yielded three different phase
speed estimates. The application of a single method over the entire range of latitudes yields estimates that occasionally vary by over 50 % between adjacent 0.5° latitudes so the latitudinal continuity of the phase speed rules out the use of a single method. In only one or two latitudes (out of 22) all three methods have yielded the same (up to 10 %) estimate. Our conclusion from these comparisons bolsters our criteria that only when at least two of the three methods yield phase speed estimates that are closer to one another than 10 %, the resulting phase speed estimate can be considered reliable.

Phase speed estimates north of 35° S and between 37° S and 39° S have not satisfied the agreement criteria between methods outlined in the end of Sect. 2.2. The lack of reliable phase speed estimates at these latitudes even though the amplitudes of the SSHA there are higher than in adjacent latitudes in which the phase speed estimates were deemed reliable (and especially north of 35° S) requires an explanation. In linear theories amplitudes can only be determined up to a multiplicative factor while phase speeds are determined completely. Accordingly, the harmonic theory where the solution is
determined by $\phi_m$ alone, does not provide any information on the variation of the SSHA amplitude with $\phi_m$, while in the trapped wave theory the variation of the amplitude with $\phi_m$ is determined up to an overall multiplicative constant. Regardless of whether the meridional structure of SSHA is determined or not it should be stressed that higher/lower amplitudes do not necessary imply that the corresponding phase speed estimates are more/less reliable and it is possible for the amplitude to be high while the phase speed estimates are not reliable (using the methods and criteria we apply) or for the phase speed to be
significant where the amplitudes are small (e.g. south of 40° S).

Figure 5 shows the observed and the two theoretical speeds as a function of $\phi_m$ between 35° S and 45.5° S where reliable estimates are obtained. The theoretical trapped speed is calculated using Eq. (4) and the theoretical harmonic speed is calculated using Eq. (2). It is clear that the trapped speeds (solid red line) are closer to the observed speeds (blue dots, squares and triangles) than the harmonic speed (dashed green line). A quantitative confirmation of this qualitative conclusion
can be obtained by calculating the sum of squares of the distances between the observed and theoretical speeds. This calculation shows that trapped speeds with sum of squares that equals 3.5 are much closer than harmonic speeds where the sum of squares is 15.3 i.e. more than four times that of trapped waves. Since the value of 10 % agreement (shown by blue circular dots) is somewhat arbitrary, we also include in Fig. 5 estimates of 11 and 12 % agreement (light-blue triangles) and estimates that agree by 25 % (light-blue squares).

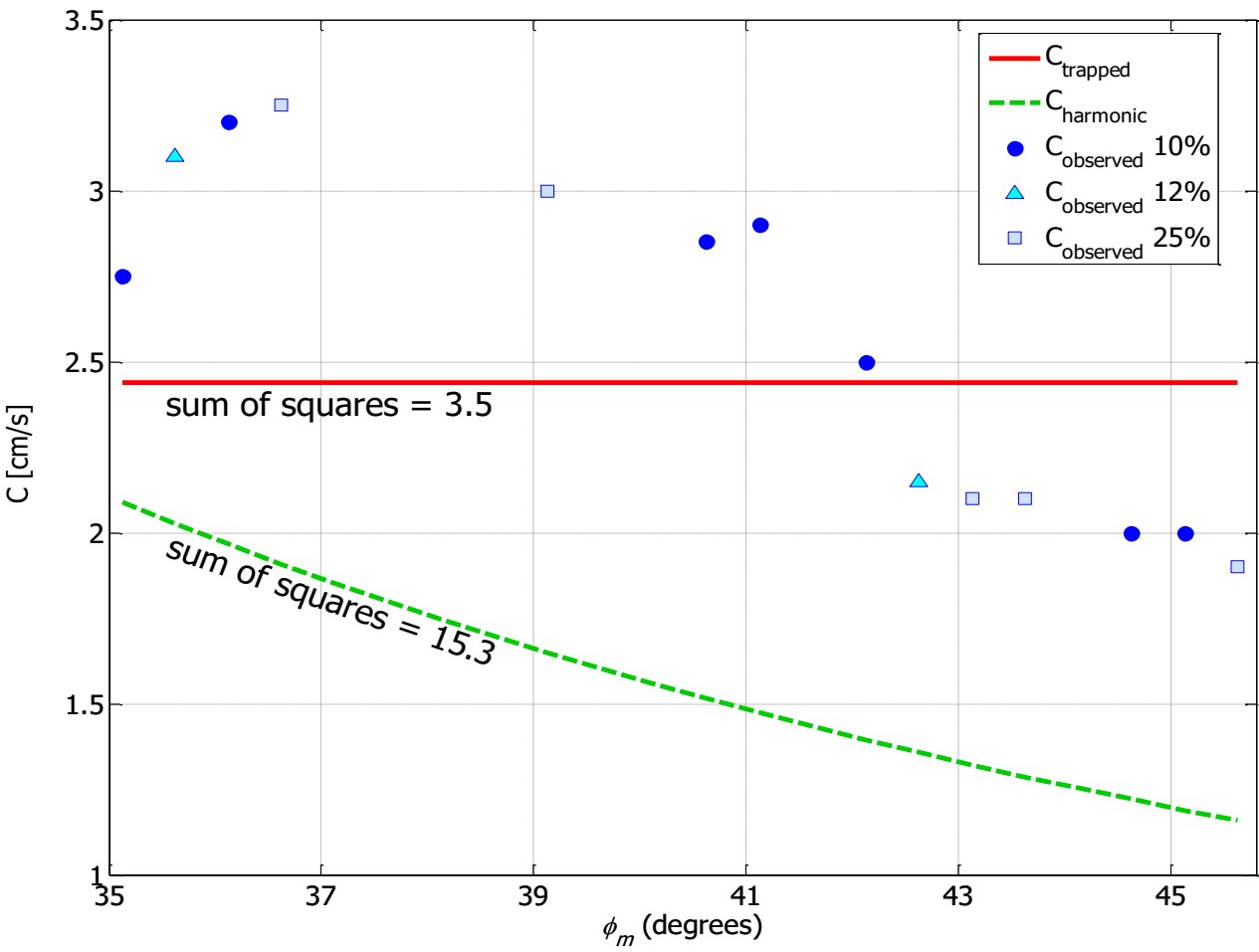

**Figure 5.** The observed phase speeds and the two theoretical phase speeds (trapped and harmonic) as a function of $\phi_m$ in intervals of 0.5° latitude. Blue dots denote latitudes where the estimates of at least two methods agreed by 10 % or less, triangles denote latitudes where such estimates agreed by 11 to 12 % and squares denote latitudes where the agreement is 25 %. No reliable estimates were obtained north of 35° S and in some more latitudes. The sum of squares of the distances in (cm s$^{-1}$)$^2$ between trapped wave phase speeds and observed speeds (3.5) is much smaller than that of harmonic phase speeds (15.3).

**6 Discussion and Summary**

The phase speed of harmonic waves decreases monotonically with the latitude of observation, $\phi_m$, as is evident from Eqs (1)-(2). In contrast, the phase speed of trapped waves depends on $\phi_w$ only (i.e. the latitude of the zonal boundary) and is independent of $\phi_m$. Our analyses of the propagation speeds of SSHA signals show that the rate at which the observed speed decreases with $\phi_m$ (the trend of the data in Figure 5 is 0.12 cm s$^{-1}$ deg$^{-1}$) exceeds the rates of decrease of both harmonic

(where the trend is 0.09 cm s$^{-1}$ deg$^{-1}$) and trapped (no trend) phase speeds. In contrast, the values of the observed phase speeds are much closer to the trapped phase speeds than to the harmonic speeds.

Colin de Verdière and Tailleux (2005) argue that the addition of mean flows affects the propagation speed of Rossby waves via its curvature: increase (decrease) of the westward phase speed for eastward (westward) surface mean flow. However, in the domain of the Indian Ocean studied here it is not clear whether or not a mean flow exists (in contrast to west of Australia where a subtropical gyre has been observed, see e.g. Stramma and Lutjeharms, 1997) and what it is its direction (some of the flows vary seasonally, see Wyrtki, 1973) so even if the numerical values of parameters such as Richardson number or buoyancy could be somehow estimated it is still unclear whether the mean flow increases or decreases the phase speed.

We should note that the simple choice, here and in many other prior studies, to interpret that observed SSHA propagation as that of the first baroclinic mode is not the only possible choice. Other choices of a single mode to fit the observation require detailed analyses of the hydrography while a (linear) combination of several modes (including the fast barotropic mode) with weights that are tuned so as to fit the observed speed can yield a better fit (see Hochet et al., 2015).

Chelton et al. (2007) and Chelton et al. (2011) argue that most of the observed SSHA features in the global ocean are nonlinear mesoscale eddies whose propagation speed is close to the phase speed of long harmonic Rossby waves (but linear eddies move much faster). The nonlinearity in those studies is determined by a combination of second order spatial derivatives of the SSHA that are used in the calculation of the Okubo-Weiss parameter. Since no derivatives can be computed by the methods in the present study it is impossible to use these methods to directly determine whether the SSHA features examined in the present study are linear or not. However, all observed propagation speeds calculated here move faster than the phase of long harmonic Rossby waves (see Fig. 5) which implies that only linear (in the sense defined in Chelton et al., 2007) eddies that propagate faster than the phase speed of long harmonic Rossby waves exist in the Indian Ocean south of Australia.

As was concluded in De-Leon and Paldor (2016) an estimation of the observed phase speed using one method only, is not reliable in most of the observed signals. On the other hand, even when estimates of the observed speed of at least two methods agree with each other, a comparison of the observed speed and the theoretical speeds varies in accordance with the method used for obtaining the observed speed. For example, in Fig. 4 the observed speed obtained by the variance method (panel b) is much closer to the trapped speed than to the harmonic speed while the observed speed obtained by the Radon and 2D FFT methods does not fit either the trapped or the harmonic speed preferentially. These differences between different methods point to the low accuracy/reliability of existing SSHA data.

As mentioned in the introduction, many studies compared observations of Rossby waves in the ocean with the harmonic Rossby waves. However from a theoretical point of view, the harmonic theory in mid-latitudes is valid only in domains narrower than a few hundred kilometres so it is not clear why one should expect the harmonic speed to match the observed

speed at unbounded domains. The case of Australia is unique since the trapped wave theory applies there while no other place exists that has a sufficiently wide, nearly straight, zonal coast line and meridional extent of the ocean that spans over 10° poleward of the equatorward boundary. In cases of narrower straight zonal coast line such as Puerto Rico the trapped wave theory is inapplicable. In unbounded domains of the world ocean the trapped wave theory does not apply straightforwardly and additional theoretical considerations have to be developed.

## Acknowledgements

The authors are grateful to C. Wunsch of MIT/Harvard University for his helpful and instructive comments on an earlier version of this work. The comments of two anonymous reviewers helped us clarify the focus of the paper and improve its presentation.

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
