# Peer review of "Trapped Planetary (Rossby) Waves Observed in the Indian Ocean by Satellite Borne Altimeters"

_Ocean Science, 2017_

## Referee Comment (RC1) · Anonymous Referee #1 · 16 Mar 2017

The manuscript presents estimates of the propagation velocity of Rossby waves at the Indian Ocean, south of Australia, using different methods, and compare them with current theories. The estimates are obtained from satellite data.

In general, the manuscript is very clear and the results present a coherence between observations and the theory. However, there are some few questions that would need more explanations regarding, mainly, the estimates close to coast. This issues follow below.

-In page 4, line 9, it is not clear why the mean ssh from 1993 was removed. Why not removing the average for the whole period? -In page 6 the authors mentioned an increase of the amplitude close to the coast. Although they state the topography is not

an issue for that, it is hard to assume it. First, close to the coast, satellite altimeter sampling is not very accurate. Secondly, and most important, that region seems to include the slope and the shelf, where other processes are very important, such as topographic waves and continental shelf waves. The main dynamics is not determined by the theory presented by the authors in that region. -In page 14 the authors did not consider other explanations, as mentioned before, associated with the topography. The same theory is applied for a region with very distinct characteristics and as such should be analyzed. I would recommend removing the analysis of the data between the coast and 35oS. -The authors present 3 different methods to estimate the propagation speed, and consider all of them with the same reliability. It would be a good contribution if the differences among those methods could be discussed and some suggestion about the method that could better estimate the propagation speed in the domain could be given. -It would also be a good contribution a discussion about which of the current theories could better explain the observed propagation velocities.

As a conclusion, I would suggest the publication of the article after addressing the issues stated above.

---

## Referee Comment (RC2) · Anonymous Referee #2 · 23 Mar 2017

The Authors use sea surface height anomaly data from satellite altimeters to identify Rossby Waves at the Great Australian Bight. They employ three different methods to estimate the phase speed of the waves, and then compare their results with both the classic harmonic theory and a more recent trapped waves theory. The study main conclusion is that the observed phenomena are Trapped Rossby Waves.

The manuscript is well written, however I have some issues with the content.

**Major reviews**

1) The trapped wave theory is very recent and I'm not sure the community is well aware of it. Since it is of utmost importance for this work, I'm not satisfied by the

provided explanation in page 9. I do not expect the authors to derive the full theory, but a more detailed explanation is due. Although, I'd like to see the explanation to start with equation 4 from Gildor et al. (2016), I think it would be appropriate if it started at least with equation 6. The first paragraph of page 9 was impossible to follow without Gildor et al. (2016) open beside it.

2) This comment is more general about the classical linear theory phase speed used in this study. Yes, it is slower than the observed one, as was found in many other studies. However, all the theoretical advances made by Killworth and Blundel (one of their papers is cited) managed to bring the theoretical linear speed closer to the observed one. Watanabe et al. (2016, Ocean Dyn.) showed that by using an effective- $\beta$ , as proposed by Herrmann and Krauss (1989, JPO), the linear theory is good enough to explain the observations at least in the tropics. Although, the tropics is not the authors' case, the main idea is that it is not a surprise that the linear theory fails to reproduce the observations, if it was calculated without considering other parameters (as in the extended theory of Killworth and Blundel, or parameterized in the effective- $\beta$ as in Watanabe et al.). Also, the linear theory is for free waves, and Rossby waves can be forced by Ekman pumping. In my opinion, there should be some discussion about these matters.

**Minor reviews**

1) Since it is a study about Rossby waves, I missed some more recent references, such as O'Brien et al. (2013, Remote Sens. Environ.) and Polito and Sato (2015, J. Geophys. Res.), among others. There is also quite a few works from Dr. Angela Maharaj that could be useful as reference. However, the reference I missed the most is Potemra (2001, J. Geophys. Res.), since it is one of the first studies of Rossby waves in the Indian Ocean.

2) The described method based on variance (page 9) is very similar to the one used by Polito and Liu (2003, J. Geophys. Res.), the only difference is that they used in
filtered data, however that does not change the method, so I think they should be cited. Also, they showed how this method is superior to the traditional Radon transform, so the relevant comparison is between the Radon transform based on variance and the 2DFFT one.

3) The explanation about the boundaries of the trapped wave theory is made in the introduction (page 3, lines 11-14). I believe it would be more appropriated if it is moved to section 4 or at least repeated there.

4) Figure 1 looks like from Google Maps. There should be some referencing, right? Ignore this comment if it was indeed rendered by Authors.

5) In page 8, the explanation of harmonic theory (line 15 until end of page) is more detailed then needed (unlike the trapped wave theory). I recommend the Authors to just present the general phase speed and the one for the long waves. In this case, to just cite the basic literature is sufficient.

6) I did not like the math notation. Since the authors are using the  $\beta$  plane, there is no reason to not use  $f_0$  and  $\beta$ . The full expression of both are cumbersome. Also, it is easier to realize what is the last term in the denominator of equation 3 (and whenever it is shown again), if it is shown as a multiplication of  $2\beta$  and  $f_0/(g'H')$ , instead of the full expression  $(2\Omega)^2 sin(2\phi_w)/(ag'H')$ . So please change accordingly.

7) Lines 11-12 of page 18: add a "in mid latitudes" after "[...] the harmonic theory is valid only in domains narrower than a few hundred kilometres [...]". In the tropics, it works for a few thousand kilometers, as shown by Watanabe et. al (2016, Ocean Dyn.).

8) In the abstract, in the penultimate sentence, please add something like "as was observed by previous studies" after the "[...] 140% to 200%". This result is not exactly unexpected, and the abstract will be better if it acknowledges this.
Other

The jet colormap is not very good. A sequential colormap for spectral amplitude and a divergent one for SSHA would be substantially better. However, since the colormap jet is well accepted by the community (no idea why), I leave it to the Authors' discretion.

**Final remarks**

I would like to add that I enjoyed reading the manuscript, and I think it is very interesting. I'm sure the Author's will have no difficult in addressing the proposed reviews, and I'll be glad to recommend it to publishing then.

---

## Author Comment (AC2) · 2 Apr 2017

Authors' detailed response to the comments in rc2 are provided in the attached file in which the original comments appear in black while the authors' responses appear red

Please also note the supplement to this comment:
http://www.ocean-sci-discuss.net/os-2017-3/os-2017-3-AC2-supplement.pdf

---

## Author Response (AR1)

Authors' response to OS-2017-03: "Trapped Planetary (Rossby) Waves Observed in the Indian Ocean by Satellite Borne Altimeters" by De-Leon and Paldor.

The comments of the two reviewers were helpful to us in clarifying the focus of the paper and improving its presentation. An acknowledgement of their input was added in the Acknowledgement section. We were also happy to read both reviewers' comments regarding the importance of the paper and their wish to see the paper published. We hope that both reviewers find the current version of the paper suitable for publication in OS. Our detailed response to the particular comments listed below in red and the changes that were implemented in the text following these comments are outlined (when relevant) in green.

**Anonymous Referee #1**

The manuscript presents estimates of the propagation velocity of Rossby waves at the Indian Ocean, south of Australia, using different methods, and compare them with current theories. The estimates are obtained from satellite data.

In general, the manuscript is very clear and the results present a coherence between observations and the theory. However, there are some few questions that would need more explanations regarding, mainly, the estimates close to coast. This issues follow below.

1) In page 4, line 9, it is not clear why the mean ssh from 1993 was removed. Why not removing the average for the whole period?
   The removal of the mean SSH of 1993 is done by Aviso (probably as part of their calibration process) Since this is a technical point that relates to the way Aviso generates the data they distribute we refer the reader to Aviso's web site for further details on the way data is produced instead of providing some of these details in our paper. Text will be modified accordingly
   See deleted sentences in the first paragraph on (new) P. 4.

2) In page 6 the authors mentioned an increase of the amplitude close to the coast. Although they state the topography is not an issue for that, it is hard to assume it. First, close to the coast, satellite altimeter sampling is not very accurate. Secondly, and most important, that region seems to include the slope and the shelf, where other processes are very important, such as topographic waves and continental shelf waves. The main dynamics is not determined by the theory presented by the authors in that region.
   Topographic Rossby waves as well as all other high-frequency waves are all filtered out by our 35 days averaging. The accuracy by Aviso data near the coast has been improved in the 20-year product used in this study (and in any case it should not be expected lead to an increase in the signal). The text will be modified to reflect this issue.
   The requested information has been added in the first paragraph on (new) P. 7

3) In page 14 the authors did not consider other explanations, as mentioned before, associated with the topography. The same theory is applied for a region with very distinct characteristics and as such should be analyzed. I would recommend removing the analysis of the data between the coast and 35oS.

Done. Figure 5 will be removed in the revised version.

4) The authors present 3 different methods to estimate the propagation speed, and consider all of them with the same reliability. It would be a good contribution if the differences among those methods could be discussed and some suggestion about the method that could better estimate the propagation speed in the domain could be given.

The interested reader can find a detailed description of the various methods, including a comparison between them in De-Leon and Paldor, 2016 in Acta Astronautica (which is referenced in the manuscript) and repeating it in the present work is an unwanted digression.

5) It would also be a good contribution a discussion about which of the current theories could better explain the observed propagation velocities.

A detailed comparison between the applicability of harmonic and Trapped theories to observations is given in (new) Figure 5 (i.e. Figure 6 of the original manuscript) and in the discussion section. A more detailed review of all the theories that have been suggested in the last 20 years and their success in explaining these particular observations is beyond the scope of this paper in which we focus on the single case where a zonal boundary exists (over a sizeable range of longitudes) where the trapped wave theory is relevant.

**Anonymous Referee #2**

The Authors use sea surface height anomaly data from satellite altimeters to identify Rossby Waves at the Great Australian Bight. They employ three different methods to estimate the phase speed of the waves, and then compare their results with both the classic harmonic theory and a more recent trapped waves theory. The study main conclusion is that the observed phenomena are Trapped Rossby Waves. The manuscript is well written, however I have some issues with the content.

**Major reviews**

1) The trapped wave theory is very recent and I'm not sure the community is well aware of it. Since it is of utmost importance for this work, I'm not satisfied by the provided explanation in page 9. I do not expect the authors to derive the full theory, but a more detailed explanation is due. Although, I'd like to see the explanation to start with equation 4 from Gildor et al. (2016), I think it would be appropriate if it started at least with equation 6. The first paragraph of page 9 was impossible to follow without Gildor et al. (2016) open beside it.

Done. Sub-section 4.1 is extended in the revised version to include more details on the trapped wave theory and the correspondence between the present work and Gildor et al (2016).

New equation 3 on P. 9 is (nearly) identical with equation 6 of Gildor et al. (2016). The discussion of the theory is now provided in the text before and after this equation.

2) This comment is more general about the classical linear theory phase speed used in this study. Yes, it is slower than the observed one, as was found in many other studies. However, all the theoretical advances made by Killworth and Blundel (one of their papers is cited) managed to bring the theoretical linear speed closer to the observed one. Watanabe et al. (2016, Ocean Dyn.) showed that by using an effective- β, as proposed by Herrmann and Krauss (1989, JPO), the linear theory is good enough to explain the observations at least in the tropics. Although, the tropics is not the authors' case, the main idea is that it is not a surprise that the linear theory fails to reproduce the observations, if it was calculated without considering other parameters (as in the extended theory of Killworth and Blundel, or parameterized in the effective-β as in Watanabe et al.). Also, the linear theory is for free waves, and Rossby waves can be forced by Ekman pumping. In my opinion, there should be some discussion about these matters.

Done. Requested references and discussions added in the revised version.
See discussion in 3$^{rd}$ paragraph on new P. 2 and the revised list of references on P. 19

**Minor reviews**

1) Since it is a study about Rossby waves, I missed some more recent references, such as O'Brien et al. (2013, Remote Sens. Environ.) and Polito and Sato (2015, J. Geophys. Res.), among others. There is also quite a few works from Dr. Angela Maharaj that could be useful as reference. However, the reference I missed the most is Potemra (2001, J. Geophys. Res.), since it is one of the first studies of Rossby waves in the Indian Ocean.

Done. Additional references and discussions added in the revised version.
See first paragraph on P. 2, paragraph above Figure 2 on P. 7 and new list of references

2) The described method based on variance (page 9) is very similar to the one used by Polito and Liu (2003, J. Geophys. Res.), the only difference is that they used in filtered data, however that does not change the method, so I think they should be cited. Also, they showed how this method is superior to the traditional Radon transform, so the relevant comparison is between the Radon transform based on variance and the 2DFFT one.

Done. Reference added in the revised version. Indeed the superiority of the variance-based Radon transform over the traditional (sum-based) counterpart in our AA paper agrees with the conclusions of Polito and Liu (2003).
See first paragraph on P. 5 and list of references

3) The explanation about the boundaries of the trapped wave theory is made in the introduction (page 3, lines 11-14). I believe it would be more appropriated if it is moved to section 4 or at least repeated there.

Done. In the revised version of the paper the explanation appears in sub-section 4.1
See first paragraph on P. 9

4) Figure 1 looks like from Google Maps. There should be some referencing, right? Ignore this comment if it was indeed rendered by Authors.

Done.
See caption of Figure 1

5) In page 8, the explanation of harmonic theory (line 15 until end of page) is more detailed then needed (unlike the trapped wave theory). I recommend the Authors to just present the general phase speed and the one for the long waves. In this case, to just cite the basic literature is sufficient.
Done. Explanation shortened in the revised version (for the most part it now includes explanations of the symbols only).
See changes in section 4.1 on P. 8

6) I did not like the math notation. Since the authors are using the β plane, there is no reason to not use f0 and β. The full expression of both are cumbersome. Also, it is easier to realize what is the last term in the denominator of equation 3 (and whenever it is shown again), if it is shown as a multiplication of 2β and f0/(g 0H0 ), instead of the full expression (2Ω)2 sin(2φw)/(ag0H0 ). So please change accordingly.
Done. The notation of $f_0$ and β has been used (but we also added the explicit expressions of these parameters since f(y) is expanded about different mean latitudes (i.e. $\phi_m$, in the harmonic theory and $\phi_w$ in the trapped wave theory).
See changes in section 4.1 on P. 8

7) Lines 11-12 of page 18: add a "in mid latitudes" after "[...] the harmonic theory is valid only in domains narrower than a few hundred kilometres [...]". In the tropics, it works for a few thousand kilometers, as shown by Watanabe et. al (2016, Ocean Dyn.).
Done.
See second paragraph on P. 17

8) In the abstract, in the penultimate sentence, please add something like "as was observed by previous studies" after the "[...] 140% to 200%". This result is not exactly unexpected, and the abstract will be better if it acknowledges this.
Done.
See new Abstract

**Other**

The jet colormap is not very good. A sequential colormap for spectral amplitude and a divergent one for SSHA would be substantially better. However, since the colormap jet is well accepted by the community (no idea why), I leave it to the Authors' discretion.

Since this is the accepted style in the community we've chosen to adhere to the existing colomap.
**Final remarks**

I would like to add that I enjoyed reading the manuscript, and I think it is very interesting. I'm sure the Author's will have no difficult in addressing the proposed reviews, and I'll be glad to recommend it to publishing then.

Thank you for the compliment.

[revised manuscript text omitted]
 first 3 values of $\xi_n$ are: $\xi_0$=1.0188; $\xi_1$=3.2482 and $\xi_2$=4.8201 (see P. 478 of Abramowitz and Stegun, 1972). The resulting expression for the phase speed of the first baroclinic mode of sufficiently long trapped waves (i.e., for zonal wavenumber $k$=0 and meridional mode number $n$=0 for which $\xi_0$=1.0188) isFollowing the studies of Paldor and Sigalov (2008), De-Leon and Paldor (2009) and Gildor et al. (2016) we expand the Coriolis frequency near $\phi_0 = \phi_w$, where $\phi_w$ is the latitude of the equatoward boundary of the domain. The results shown in Fig. 2(b) clearly indicate that the meridional derivative of the height field vanishes at $\phi_w$. Formally, the expression for the phase speed in the present set-up is obtained by setting $L$=0 in Eq. (6) of Gildor et al. (2016) (the reason is that in the latter study the wall was placed at $y=-L/2$ while in the present study the wall is located at $y=0$). In addition, the boundary condition at $\phi_w$ in the present application differs from that in Gildor et al. (2016) in which the boundary condition was the natural condition of the vanishing of the meridional velocity at $\phi_w$. Instead, as shown in Fig. 2(b) in the present study the meridional derivative of the height field vanishes at $\phi_w$. Thus, $\xi_n$ in Eq. (6) of Gildor et al. (2016) (which is the absolute value of the $n$th zero of $Ai$ - the regular Airy function) should be replaced in the present application by $\zeta_n$ (where $\zeta_n$ is the absolute value of the $n$th zero of $Ai'$, the derivative of $Ai$). The first few values of $\zeta_n$ are: $\zeta_0$=1.0188; $\zeta_1$=3.2482 and $\zeta_2$=4.8201 (see P. 478 of Abramowitz and Stegun, 1972). 
[revised manuscript text omitted]

[Figure]

**Figure 4. Objective analysis of the phase speeds associated with the time-longitude diagram of Fig. 3(b). (a) The distribution of the sum-of-squares of the Radon transform versus** $\theta$ **(near the peak) normalized such that the maximum value equals 1 (solid blue curve) and the two angles corresponding to the phase speeds of the harmonic wave theory (Eq. (2); dashed green vertical line) and to the trapped wave theory (Eq. (4); solid red vertical line). Note that the same two vertical lines appear also in panels (b) and (d).**

5 **(b) The distribution of the mean of variances versus** $\theta$**, normalized such that the maximum value equals 1 and the minimum is 0 (solid blue curve). (c) The 2D FFT frequency-wavenumber diagram in the low frequency-low wavenumber regime (where the wavenumber** $k$ **is measured in units of (¼ degrees of longitude)$^{-1}$, the frequency** $\omega$ **is measured in units of week$^{-1}$ and the amplitude is calculated in arbitrary units). Dashed red line: phase speed of the trapped wave theory (Eq. (4)). Dotted light-green line: phase speed of harmonic wave theory (Eq. (2)). The line corresponding to trapped wave connects the two maximal values of the 2D FFT**

10 **at the smallest** $\pm k \neq 0$ **which also passes through the origin but does not follow the location of maximal amplitude at larger wavenumbers better than the line corresponding to harmonic waves. (d) The distribution of the sum-of-squares of the 2D FFT amplitudes along different lines (sweeping) versus arctan(**$C$**), normalized such that the maximum value equals 1 (blue curve). Only values of arctan(**$C$**) in the range of 90°-180° are shown since only these values correspond to westward propagating phase speed.**

15 The implications from similar comparisons carried out every 0.5° between 33° S and 45.5° S can be summarized as follows: At about third of the diagrams analyzed the signal was too blurred or the three methods yielded three different phase speed estimates. The application of a single method over the entire range of latitudes yields estimates that occasionally vary by over 50 % between adjacent 0.5° latitudes so the latitudinal continuity of the phase speed rules out the use of a single method. In only one or two latitudes (out of 22) all three methods have yielded the same (up to 10 %) estimate. Our

20 conclusion from these comparisons bolsters our criteria that only when at least two of the three methods yield phase speed estimates that are closer to one another than 10 %, the resulting phase speed estimate can be considered reliable.

Phase speed estimates north of 35° S and between 37° S and 39° S have not satisfied the agreement criteria between methods outlined in the end of Sect. 2.2. The lack of reliable phase speed estimates at these latitudes even though the amplitudes of the SSHA there are higher than in adjacent latitudes in which the phase speed estimates were deemed reliable

25 (and especially north of 35° S) requires an explanation. In linear theories amplitudes can only be determined up to a multiplicative factor while phase speeds are determined completely. Accordingly, the harmonic theory where the solution is determined by $\phi_m$ alone, does not provide any information on the variation of the SSHA amplitude with $\phi_m$, while in the trapped wave theory the variation of the amplitude with $\phi_m$ is determined up to an overall multiplicative constant. Regardless of whether the meridional structure of SSHA is determined or not it should be stressed that higher/lower amplitudes do not

30 necessary imply that the corresponding phase speed estimates are more/less reliable and it is possible for the amplitude to be high while the phase speed estimates are not reliable (using the methods and criteria we apply) or for the phase speed to be significant where the amplitudes are small (e.g. south of 40° S).

Figure 5 shows the observed and the two theoretical speeds as a function of $\phi_m$ between 35° S and 45.5° S where reliable estimates are obtained. The theoretical trapped speed is calculated using Eq. (4) and the theoretical harmonic speed is

35 calculated using Eq. (2). It is clear that the trapped speeds (solid red line) are closer to the observed speeds (blue dots,

¶
¶
¶
¶
¶
<sp>¶
<sp><sp><sp>

¶
¶
¶
¶
¶
¶
¶
¶

squares and triangles) than the harmonic speed (dashed green line). A quantitative confirmation of this qualitative conclusion can be obtained by calculating the sum of squares of the distances between the observed and theoretical speeds. This calculation shows that trapped speeds with sum of squares that equals 3.5 are much closer than harmonic speeds where the sum of squares is 15.3 i.e. more than four times that of trapped waves. Since the value of 10 % agreement (shown by blue circular dots) is somewhat arbitrary, we also include in Fig. 5 estimates of 11 and 12 % agreement (light-blue triangles) and estimates that agree by 25 % (light-blue squares).

**Moved down [2]:** In some latitudes no observed values are given since they have not passed the criteria outlined above for determining the observed phase speed (i.e. an agreement between two or more of the various methods).

[revised manuscript text omitted]